# Teleporting Windows to Improve Virtual Reality Cockpits

Category: Research

## ABSTRACT

Virtual reality allows a user to be surrounded by multiple floating virtual screens or windows in a kind of cockpit. Such arrangements increase the user's ability to monitor several windows and quickly switch between windows with rapid eye motions. However, users sitting in a physical chair may suffer from neck fatigue if they fixate an off-center window for long periods of time. We propose an interaction technique for quickly teleporting any window to the center of the workspace, allowing the user to fixate that window with their neck in a neutral position, and later dismiss the window which snaps back to its original position. We present an experimental comparison of (1) a Cockpit whose windows are fixed, (2) a Cockpit with Teleportation, allowing any window to be temporarily moved to the center, and (3) a condition similar to a single Desktop screen where only one window is visible at a time, and a key combination similar to Alt+Tab switches between windows. Teleportation resulted in less neck fatigue than Cockpit, as subjectively reported by users, and was also preferred by a majority of users. We subsequently present a prototype that sketches how to extend teleportation to multiple simultaneous windows.

**Index Terms:** Human-centered computing—Virtual reality; Human-centered computing—Graphical user interfaces

## 1 INTRODUCTION

There is growing interest in academia [11, 12, 24, 34] and industry [27] to use virtual reality (VR) headsets and similar technologies to surround a user with multiple virtual desktops or windows. Tens of thousands of users already regularly view and interact with their PC's virtual desktop in VR[1]. As the resolution of headsets continues to improve, the increased real estate available in a VR "cockpit" (Figure 1) could allow a user to monitor greater amounts of information [13, 16, 22], quickly switch their attention between applications by simply rotating their eyes and/or neck, and also shield the user from external distractions.

One disadvantage of a virtual cockpit is that the user may suffer from neck fatigue if they must attend to an off-center window for more than a few moments. If the user is engaged in knowledge work, they are likely seated and using a physical keyboard and mouse, making it difficult to rotate their body to face the current window.

We therefore designed an interaction technique, called Teleportation, allowing the user to temporarily move any window to the center of their workspace for maximal comfort. The teleported window "remembers" its original position, to which it returns as soon as the user dismisses the window or teleports a different window to the center. Thus, the original layout of the windows is preserved, allowing the user to leverage their spatial memory to find windows. The user may rotate their head and move their mouse cursor to any window for a brief glance or interaction, or optionally invoke teleportation for more prolonged and comfortable work with any given window.

Our contributions are (1) the design of the teleportation interaction technique, (2) a simple set of 4 tasks for simulating multi-window work that can be used to evaluate future window management techniques, and (3) the results of a controlled experiment that used the 4 tasks to compare three user interface conditions: a normal Cockpit where windows have fixed positions, a "Cockpit with Teleportation" where windows can be moved to the center, and a "Desktop"

---

[1]Guy Godin's VRDesktop "used by more than 40k people every day" https://twitter.com/VRDesktop/status/1359606565808926723

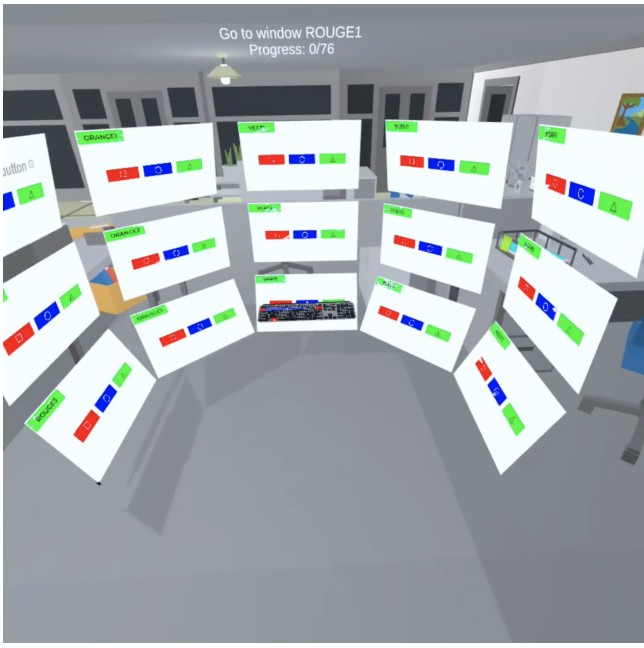

Figure 1: In our cockpit, the user is surrounded by 15 windows, with a virtual keyboard (visible in the bottom row, middle column) indicating the physical keyboard's location. Not shown here is the chair where the user sits.

condition where only one window is visible at a time and a key combination similar to Alt+Tab is used to switch between windows. Out Of 16 participants, 9 chose the Cockpit with Teleportation as their favorite of the three conditions. Cockpit with Teleportation also resulted in less neck fatigue than the normal Cockpit, as subjectively judged by users. We also present (4) a prototype that sketches how to extend teleportation to multiple simultaneous windows.

## 2 BACKGROUND

We review previous work on window management, first for 2D virtual desktops, and then for immersive 3D environments in VR and augmented reality (AR).

### 2.1 Window Management on 2D Desktops

On 2D desktops, previous work has proposed interaction techniques to make it faster for a user to configure the size or position of a window, either manually [2, 15] or automatically [5]. Other work has proposed ways to make it easier for the user to retrieve "tasks" (i.e., subsets of windows), or subsets of documents, where these subsets might be defined manually [1, 4, 23, 33, 38] or deduced automatically [6, 28, 29, 42]. Other previous systems present the user with a spatial layout of thumbnails of windows from which the user can retrieve one or more windows. These layouts might be defined manually [33, 40] or computed automatically [6, 39]. If the layout is stable over time, the user can leverage their spatial memory to remember where a window or set of windows is located, making retrieval faster.

Although a notion of subsets can make it easier to restore a group

of windows, as pointed out by Tak et al. [39], it can be inconvenient to require every window to be part of a subset, or to prevent a window from being a member of multiple subsets. In addition, the automatic computation of subsets or of layouts always implies a risk of producing output that is undesirable or confusing to the user. Our work does not involve retrieval of subsets of windows, nor does it automatically generate layouts for retrieval. Our work is somewhat comparable to WindowScape [40] or the "Exposé" feature in macOS (subsequently renamed as part of "Mission Control"): in both of those systems, the user can access an overview of minitiarized windows, and select one to be moved to the "foreground" in its full size. However, in our system, the overview space is a wide field-of-view (FoV) cockpit where windows are already at full size and have no need to be minitiarized, and indeed are fully usable and more easily recognized than thumbnails, and the "foreground" space is a narrow FoV where the user can work with the window for a prolonged period of time more easily, with little or no neck rotation. Unlike some previous work [6, 39], in our system, there is no need to have a separate layout of windows for retrieving them, since the wide FoV already has a layout that the user is presumably familiar with, making it easy for the user to find whatever window they want to "promote" (i.e., teleport) to the "foreground" (narrow FoV).

## 2.2 Window Management in VR/AR

There is more recent work on management of windows in immersive 3D spaces.

Ens et al. [9] present a design space: windows can be egocentric or exocentric; fixed or movable; far, near, or on the body; work with direct or indirect input; be tangible (i.e., mapped to a physical surface that is touched) or intangible ('in air'); have high, intermediate, or low visibility; and be positioned on a discrete grid or within a continuous space. Our work is relevant to a knowledge worker seated at a physical desk surrounded by windows, using a mouse and keyboard, hence an egocentric, fixed arrangement, using indirect input (e.g., mouse or head-gaze), and windows with high visibility. In our experiment (Section 4), windows have discrete positions, but our subsequent prototype (Section 6) allows for continuous positioning.

There are algorithmic approaches that automatically position windows in 3D or adapt to the physical environment [10, 18, 20, 26]. Projective Windows [19] is a technique for quickly positioning an individual window, but with no functionality for easily dismissing a window to return it back to a previous location. The Personal Cockpit [11] uses a window layout similar to that in our work, but it was designed for mobile use, direct input, and had a field-of-view (FoV) limited to 40 degrees to simulate commercially available augmented reality headsets. We focus on a user who is seated at a desk with a FoV of ≈90° or more.

The Bring2Me [3] system is designed for AR platforms with a narrow FoV (prototyped with a headset with 23 degree vertical FoV). Bring2Me allows the user to leverage their spatial memory to summon a menu from a known spatial location, outside the visible FoV, bringing it temporarily into the FoV in front of the user for a single selection before the menu automatically returns to its original location. Three variants of the technique are proposed, including two named TeleHead and TelePad, for the idea of teleporting the menu in front of the user. In our work, however, we teleport an entire window in front of the user for several interactions, not just a single selection within a menu, and in Section 6 we show how to extend our system to allow multiple windows to be teleported in front of the user at the same time.

## 3 Design of the Teleportation Technique

In status-quo workstations (without using a headset), the user often has a single physical screen, only one full-screen window is visible at a time, and a shortcut key (Alt+Tab on Microsoft Windows or Command ⌘+Tab on macOS) is often used to switch between windows. One advantage of such an interface is that the neck is always in a neutral position. However, switching between windows can be slow and confusing when there are many windows [39].

In contrast, a Cockpit-style interface [11] (Figure 1), where all windows are visible, allows a user to leverage their spatial memory, possibly enabling faster switching between windows since a simple neck rotation suffices, but at the cost of inducing neck fatigue.

We sought a way to combine the advantages of both of these interfaces, and took inspiration from kinesthetically-held modes [35] (also called spring-loaded modes [14] or quasimodes [32]), and also from "spring-loaded glances" [30]. We would like a way for the user to temporarily bring ("teleport") any window to the center of the workspace, and make it easy to dismiss the window so that it returns to its original position.

Selecting the window to teleport could be done with head orientation (head-gaze), eye tracking (eye-gaze), or with the mouse cursor. We decided to use head-gaze for two reasons. First, we reasoned that if a user wishes to teleport a window, they would first look at it, causing the head-gaze to already be directed (or nearly directed) toward the appropriate window, prior to moving the mouse cursor to the window. Secondly, eye-gaze would presumably also precede motion by the mouse, however tracking of eye-gaze is still not available on many headsets, and there is some evidence that head-gaze works better than eye-gaze [31].

In our system, the window currently selected by the head-gaze raycast is shown with its border in red (Figure 3, bottom). Once the desired window is highlighted in red in this manner, tapping a single key (in our prototype, the Alt key) causes that window to teleport to the center, with a smoothly animated transition at a constant speed that lasts ≈250-550 ms depending on the distance travelled. Importantly, if the mouse cursor was in the window prior to it being teleported, the cursor is "carried" by the window so that it is still inside the window when the window completes its teleportation. To dismiss the teleported window back to its original position, the user may hit the Backquote ` key, causing another animated transition. The teleported window is also automatically dismissed whenever a different window is teleported to the center, triggering two simultaneous animations. Thus, at most one window can be teleported at a time. The limitation to one teleported window at a time, and the use of animated transitions, makes the interaction technique easier to understand by new users.

The raycast used in head-gaze is not oriented straight forward. Instead, it is tilted downward by 15°, an angle chosen based on [41] which suggests that a physical monitor's center should be "15° to 25° below gaze inclination". We indeed found that this downward tilt made the selection feel somewhat more natural.

## 4 Experiment

Our goal with this experiment was to investigate the tradeoffs between different user interfaces for managing multiple windows in VR.

## 4.1 Three Main Conditions

Our experiment compared 3 main conditions: Desktop, Cockpit, and Cockpit with Teleportation. In all conditions, there were 15 windows, with unique text labels in their upper left corner to identify each window. All windows were the same size.

In the Desktop condition (Figure 2), only one window was visible at a time. Holding down the Alt key displays a menu of window names, ordered by recency. Tapping the Backquote key selects a new window, which is brought to the front when Alt is released. (We did not use Alt+Tab in our system due to a limitation with our software framework.) In the Cockpit condition (Figure 3, top), all windows were visible in 3 rows × 5 columns, and their positions were fixed. In the Cockpit with Teleportation condition (Figure 3, bottom), any

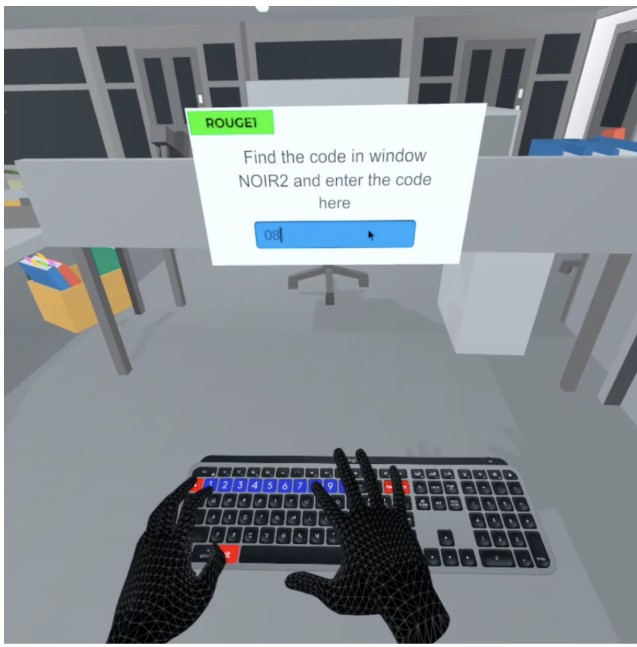

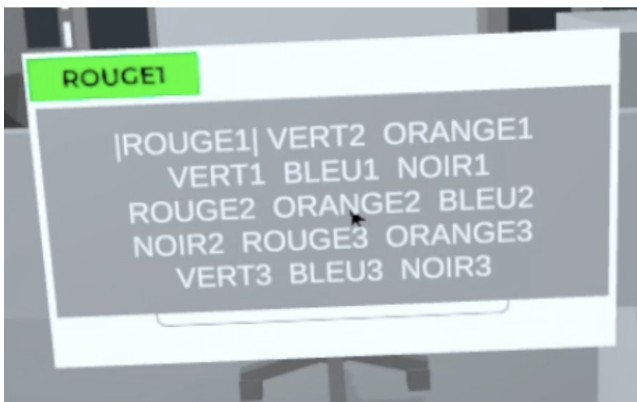

Figure 2: (Top) In the Desktop condition, all 15 windows are stacked together, with only the front-most window visible. (Bottom) Alt+Backquote on the keyboard cycles through windows, very similar to Alt+Tab on Microsoft Windows.

window could be moved to the center using the Alt and Backquote keys as described in section 3.

The tasks performed in the experiment required users to navigate between windows, with textual instructions telling the user which window to navigate to next. To imitate a scenario where the user has had enough time to learn the positions of the windows in a Cockpit, and can leverage their spatial memory, we labeled the windows with names that are easy to associate with spatial positions. Each window's label is of the form "ColorNumber", where the color corresponds to a column (from left to right, the columns are associated with labels red, orange, green, blue, or black) and the number indicates the row (1, 2, or 3). These labels were displayed in French (e.g., ROUGE2 or NOIR3), as were all instructions, as our users were francophone. In the Desktop condition, the use of text labels for distinguishing windows when using Alt+Backquote (Figure 2, bottom) is not so different from Microsoft Windows, where the thumbnails of windows are often visually similar and the user must read the names of windows while using Alt+Tab.

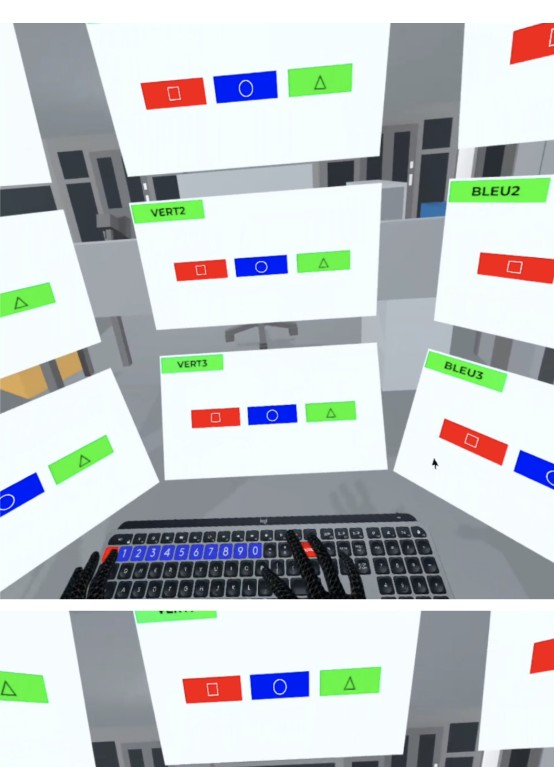

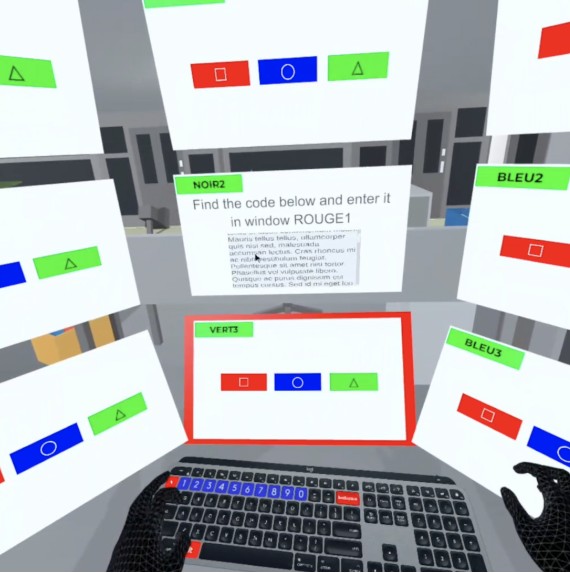

Figure 3: (Top) In the Cockpit condition, the 15 windows (in 3 rows × 5 columns) have fixed positions. (Bottom) In the "Cockpit with Teleportation" condition, any window (such as NOIR2 in this example) may be temporarily brought to the center, covering the VERT2 window. Even though the mouse cursor is in the NOIR2 window in the center, the VERT3 window in the bottom row has its border in red, to show that it is being selected by the head-gaze raycast.

### 4.2 Hypothesis

**We hypothesize that** Cockpit will result in more neck fatigue than Desktop, that Desktop will require more time than Cockpit to switch between windows, and that Cockpit with Teleportation will yield a compromise between the other two conditions in terms of time and neck fatigue.

### 4.3 Size and positioning of windows

All windows were 24 inches diagonally, with a 16/9 ratio ($\approx$ $53.1 \times 29.9$cm), to imitate full-screen windows on a common size of physical monitors.

In the Desktop condition, the window was 80cm in front of the

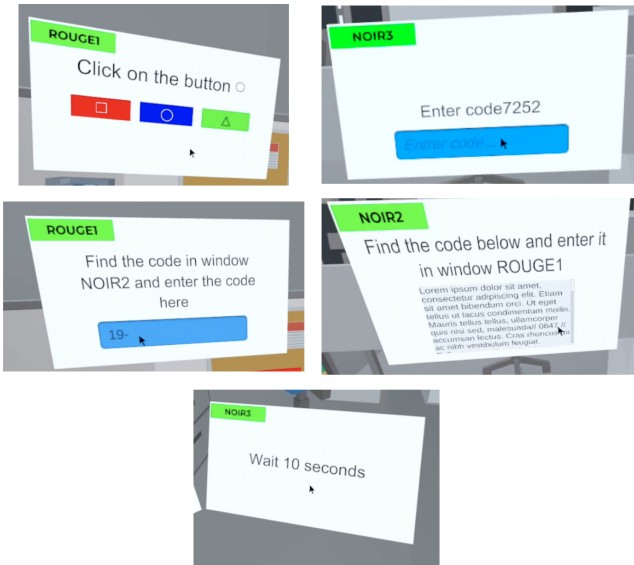

Figure 4: Top left: the Click task. Top right: the Write task. Middle row: the Read task. Bottom: the Wait task.

user's head's initial position (a distance within published recommendations [36, 41]), oriented vertically, with its center slightly lower than the user's head's initial position, so that a line connecting the two formed a 15° angle with the horizontal [41].

In the Cockpit and Cockpit with Teleportation conditions, the windows in the middle row are vertically oriented, and their centers follow a circular arc of 80cm radius and span 150°, at the same height as in the Desktop condition (so the centers are 15° down with respect to the user's head's initial position). The windows in the top row have the same horizontal positions as the middle row, but are vertically higher, and are tilted to be directly facing the user's head's initial position (i.e., no 15° tilt). In the bottom row, the centers of windows follow a circular arc of 72cm radius, and the windows are tilted to again create a 15° tilt with respect to the user's head's initial position. The 72cm radius of the bottom row is small enough to avoid occlusion from windows in the middle row, while also being big enough to avoid overlap between windows of the bottom row.

At a distance of 80cm, each window covers a horizontal angle of $2 \operatorname{atan}(53.1cm/2/80cm) \approx 36.7°$. If the neck is rotated to center the head on a given window, the eyes can fixate any point on the window by yawing less than 20° with no further neck rotation. Previous studies of neck and eye rotation [37] indicate that such shifts in gaze would entail very little neck rotation. Thus, we expect the maximum rotation (yaw) necessary by the neck in our Cockpit layout to be $150°/2 = 75°$ in either direction.

### 4.4 Four Tasks

To design our tasks, we took inspiration from two sources. First, Liu et al. [21] defined a simple, parametrized task that can be adjusted to different levels of difficulty, requiring varying amounts of zooming or panning, and that has been reused in subsequent work [25], but that is more relevant to large or zoomable displays than multi-window arrangements. Second, Ens et al. [11], in their "Study 4", aimed for an "ecologically valid task" and created a set of "everyday applications" (Contacts, Calendar, Map, etc.) where the user had to switch between windows. However, such applications are not easy to re-implement in future studies, and seem less amenable to algorithmic generation of sequences of tasks. We combined these two approaches with a set of tasks that are simple to implement and simple to explain to a user, that involve actions used in real knowl-

edge work, that require using a keyboard and mouse and multiple windows, and that can be parametrized and reused in future studies involving multiple windows.

We settled on the following 4 tasks (Figure 4). In the **Click** task, a textual message in a window tells the user which of a set of buttons (in the same window) to click with the mouse. In the **Write** task, a textual message tells the user what string (a 4-digit code) to type into a text field (in the same window). The user gives keyboard focus to the text field by clicking on it. In the **Read** task, a textual message in one window tells the user to go to another window where they must scroll (using the mouse) within a long text of words, looking for a 4-digit numeric code, and once they find that code, they must return to the initial window to type the code into a text field. In the **Wait** task, the user clicks on a button to start a 10-second countdown timer and must wait for the timer to finish (this simulates watching a video, or an ad, or waiting for a software job to complete). All 4 tasks require the user to read text and use the mouse; the Write and Read tasks also require the use of the keyboard; and the Read task requires navigating between two different windows. In the first 3 tasks, if the user clicked on the wrong button or typed the wrong string, visual feedback indicated their error, and they had to click on the correct button or use backspace to correct the string before the task was considered complete.

In addition, a *sequence* of tasks can be defined that involve different windows, sometimes requiring the user to navigate to a different window before starting the next task. Our system identifies each window with a label in the window's upper-left corner, and tells the user which window to navigate to for the next task with a textual message in the previously active window, as well as with a global message displayed above all the windows.

There are at least three parameters that can be manipulated in defining such a sequence of tasks: the distance between consecutive windows (requiring the user to navigate short or long distances), the number of consecutive tasks to do in the same window before asking the user to switch to a different window (manipulating this changes the frequency with which the user must change windows), and the total number of windows to be managed and navigated between. As the number of windows increases, or the distance between consecutive windows increases, and as the frequency of switching decreases, we expect neck fatigue to become more of a problem for cockpit-style layouts, and therefore expect teleportation to become more beneficial.

After a pilot study, we adjusted our sequences of tasks to induce more of a difference between the Desktop and Cockpit conditions, to better understand if and how Teleportation results in a compromise between these two conditions. Thus, the tasks in our sequence involve windows chosen in the left-most and right-most columns of the 3 row × 5 column layout.

### 4.5 Equipment

Each user wore a (first generation) Oculus Quest VR headset, which is untethered and performs inside-out tracking. The headset has two screens at 1440×1600 pixels each, a mass of 571 g, and a FoV of at least 90° both horizontally and vertically (the precise FoV depends on the measurement method).

A "Logitech Mx Keys" keyboard and "Logitech Mx Anywhere 2s" mouse were connected to the headset via Bluetooth. The keyboard was attached to a cardboard template with holes to insert the "Oculus Touch" controllers which are tracked by the headset (Figure 5). This allowed the user to slide the keyboard to a comfortable position at the start of the experiment, while allowing the headset to detect the keyboard's approximate position (fine tuning of this position was also done by pressing keys during a brief calibration phase). The controllers were not used during the experiment tasks. A virtual 3D model of the keyboard was displayed in VR, with labels on the keys in a large font for easier reading, and certain keys (in particular, Alt

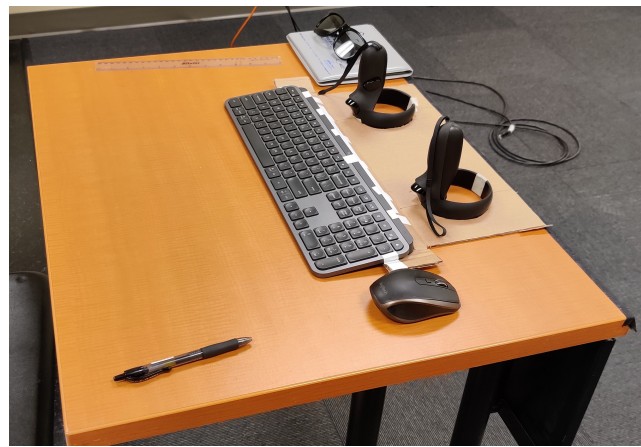

Figure 5: The controllers, normally held in the user's hands, are placed in a cardboard template, allowing the headset to deduce the location of the physical keyboard.

and Backquote) colored to remind the user of the keys to use during the experiment. No virtual model of the mouse was displayed, as this would have prolonged the calibration phase for each user, and in practice users seemed to easily remember the physical mouse's location.

Users sat on a chair that did not swivel or rotate.

The headset pushed notifications wirelessly to a computer, allowing the researchers to monitor each user's progress.

### 4.6 Participants

16 users were recruited from university engineering programs, ranging in age from 20 to 26 (mean 22.9), 3 female and 13 male, 2 left-handed and 14 right-handed, all accustomed to using the mouse in their right hand. Their depth acuity was measured with a Titmus stereo acuity test, yielding scores from 0/10 to 10/10 (mean 6.4). Only one user reported having color deficient vision. Users were asked if they had ever used multiple monitors on a regular basis; 9 answered that they had experience using 2-monitor setups, and another 3 answered that they had used 3-monitor setups. The interpupillary distance of each user was measured and used to calibrate the headset prior to the tasks.

### 4.7 Design of Experiment

Each user performed tasks in each of the 3 conditions. The ordering of conditions was random for each user. Prior to the experiment, we generated and saved 3 *sequences* of trials of tasks. For each user, these sequences were assigned at random to the 3 conditions.

For each condition, the user performed the sequence of trials in 3 phases. Phase 1 was for "warming up", to get familiar with the user interface, and involved 16 trials: 4 trials of each of the 4 types of tasks in an order determined by the pre-generated and randomly assigned sequence. Users were instructed to take their time in phase 1 and to ask questions to ensure that they understood how the interface worked. Users were asked to complete the trials in the subsequent phases faster. Phases 2 and 3 each involved 30 trials: 8 trials of Click, 8 trials of Write, 10 trials of Read, and 4 trials of Wait, but not in that order, rather in the order determined by the sequence. The trials in phase 2 were chosen so that the user would need to navigate to a new window every 2 trials, i.e., they needed to frequently change windows. The trials in phase 3 were chosen so that the user would need to navigate to a new window every 10 trials. In our results, we only present the data collected from phases 2 and 3, to cover a range of behavior (frequent and infrequent window switching) while excluding warm-up trials.

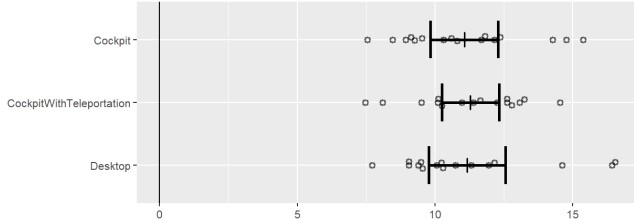

Figure 6: Average time spent per trial, in seconds.

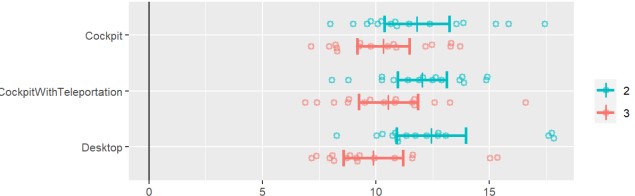

Figure 7: Average time spent per trial, in seconds, broken down by phase. (Phase 1 involved warm-up trials and is excluded.)

Throughout all phases, textual instructions in the current window and above the windows prompted the user on what to do, step-by-step. Between phases, the user was asked to remove the headset to take a break and to evaluate the cognitive load and neck fatigue of the preceding phase on Likert scales on a questionnaire.

In total, there were 16 users × 3 conditions (Desktop, Cockpit, Cockpit with Teleportation) × 2 phases (phase 2 + phase 3) × 30 trials (8 Click + 8 Write + 10 Read + 4 Wait) = 2880 trials, not including the warm-up trials performed in phase 1.

### 4.8 Results

For these results, advice was adopted from Dragicevic [8]: we avoid emphasizing null hypothesis significance testing (NHST) to avoid misleading, dichotomous thinking ("Tip 25" in [8]); we present effect sizes visually and with confidence intervals (Tips 15, 16), where the confidence intervals are computed using only one (averaged) value for each (user, condition) pair (Tip 9); and we clearly distinguish between our pre-experiment hypotheses and post-hoc exploratory data analysis, because not doing so encourages HARKing (Hypothesising After the Results are Known) and *p*-hacking.

In Figures 6-11, the error bars show 95% confidence intervals (CIs), calculated using 16 points each, i.e., one point for each user. CIs were calculated using the *t* distribution for time and angles (Figures 6-9) or using percentile bootstrapping for Likert scales (Figures 10-11).

Our hypothesis (section 4.2), was only partially confirmed. Figure 11 does show evidence that teleportation results in intermediate levels of neck fatigue, but Figure 6 does not reveal any difference in time.

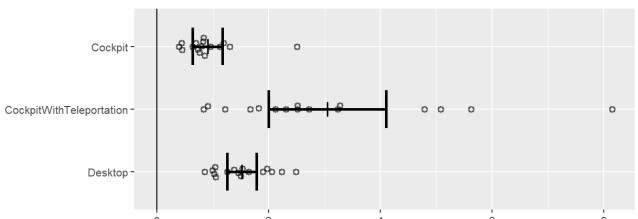

Figure 8: Estimated time spent on each navigation between windows, in seconds.

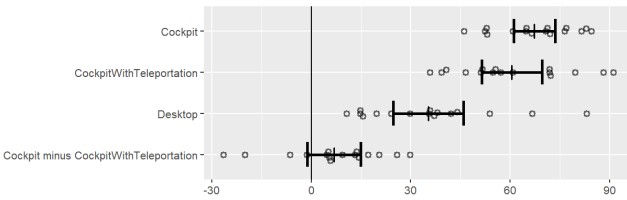

Figure 9: The standard deviation, in degrees, of the orientation of the user's head. The difference shown between two of the conditions corresponds to $p \approx 0.09$.

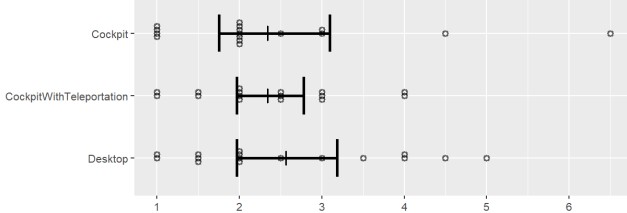

Figure 10: Cognitive load rated subjectively by users on a scale from 1 to 7 at the end of each phase of each condition.

The rest of the results presented here are post-hoc exploratory analysis.

To obtain an estimate of the time spent navigating between windows, we reanalyzed our data logs. In the Desktop condition, we counted how much time the Alt key was held down. In the other two conditions, we counted how much time the mouse cursor was not in the appropriate window for the current task (or appropriate window*s*, in the case of a Read task). We then divided this time by the number of times that the user had to change windows in each phase, yielding the estimate in Figure 8.

In the Cockpit with Teleportation condition, one of the users never used teleportation, despite being encouraged to do so. Each of the other 15 users had a window teleported during at least 83% of trials.

Figure 9 was generated by computing the angle between the user's head's "forward vector" and the workspace's fixed "forward vector" for each frame, and then finding the standard deviation of these angles for each phase of the experiment.

In post-questionnaires, users described the cockpit layout as good for seeing an overview (1 user) or that they liked the 15 windows (1 user). Teleportation was liked (4 users), easy to learn (1 user), much easier than desktop (1 user), and fun (1 user). Users asked for a cockpit layout covering a smaller FoV or with fewer windows (4 users). They also suggested that the selection of the window to teleport could be done with the mouse (3 users) or with the keyboard (1 user). Other suggestions were to combine teleportation and the Alt+Tab-style functionality (1 user), to have a physically rotating workstation where the chair, keyboard, and mouse could all rotate (1

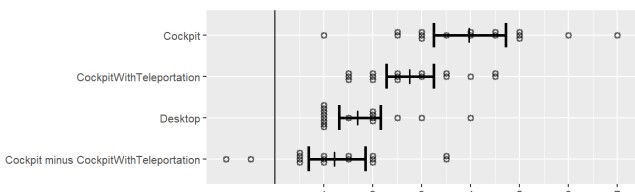

Figure 11: Neck fatigue rated subjectively by users on a scale from 1 to 7 at the end of each phase of each condition. The difference shown between two of the conditions corresponds to $p < 0.002$.

user), and to make the windows further from the center be smaller (1 user).

A majority of users preferred Cockpit with Teleportation over the other two conditions (Table 1).

Table 1: Subjectively preferred conditions by the 16 users.

| Condition | Most Favorite | Least Favorite |
|---|---|---|
| Cockpit | 1 | 12 |
| Desktop | 6 | 4 |
| Cockpit with Teleportation | 9 | 0 |

## 5 DISCUSSION

The estimated time to navigate between windows (Figure 8) was highest when using teleportation. In our own testing of the prototype, selecting the window to teleport using head-gaze was not always fast and natural: one of us often had to tilt our head downward to cause the correct window's border to highlight in red before hitting the Alt key, suggesting that the raycast was tilted too high for at least some users. At the same time, it often seemed natural to move the mouse cursor onto whatever window we attended to, even prior to teleporting a window. This suggests that the mouse cursor might be faster than head-gaze for selecting the window to teleport, and such a modification was suggested by 3 users.

Figures 9 and 11 show that, although teleportation reduced neck rotation, users still felt some fatigue. Limiting the cockpit to a smaller number of windows was suggested by multiple users. A 3×3 layout (for example) would reduce the available display space by 40%, but might indeed be preferred by users who are seated and using a keyboard and mouse. A compromise would be to have windows beyond the central 3 columns that are smaller (covering a smaller angle) and/or used mostly for monitoring data.

## 6 PROTOTYPE FOR MULTIPLE FOREGROUND WINDOWS

In real tasks, users sometimes need to frequently switch between 2 or 3 windows, to visually compare content or copy information between windows. This was partially simulated in our experiment with the Read task that always involved 2 windows, however the prototype in our experiment only allows one window to be teleported to the front at a time. We subsequently designed a 2nd prototype that allows multiple windows to be teleported in front of the user at once. This 2nd prototype was not implemented in VR, but instead is implemented with JavaScript in an HTML document, showing the windows of the cylindrical cockpit unwrapped onto a flat surface. This allowed for faster prototyping, and allows readers to try the prototype for themselves (see supplemental material). The aspect ratio of this prototype was chosen to reproduce the amount of space available to users in our experiment, with room to tile 3 rows × 5 columns of windows, each with a 16/9 aspect ratio. In line with the observations from our experiment, we assume that this prototype will only use mouse input for selecting windows to be teleported, and has no need for head-gaze input.

In our previous prototype, whenever the user teleported a new window to the center, the previously teleported window was automatically dismissed and returned to its original location. In the new prototype, the user may teleport as many windows as they wish. These teleported windows are added to a foreground layer that is automatically centered by the system. In the following description, the foreground layer refers to the subset of windows centered in front of the user within a narrow FoV, while the background layer refers to complete set of windows covering a wide FoV.

We assume that one modifier key on the keyboard can be dedicated to this new window managing approach. In our case, we use the Ctrl key for this, so that our JavaScript implementation can intercept press and release events, however in a real system, another key

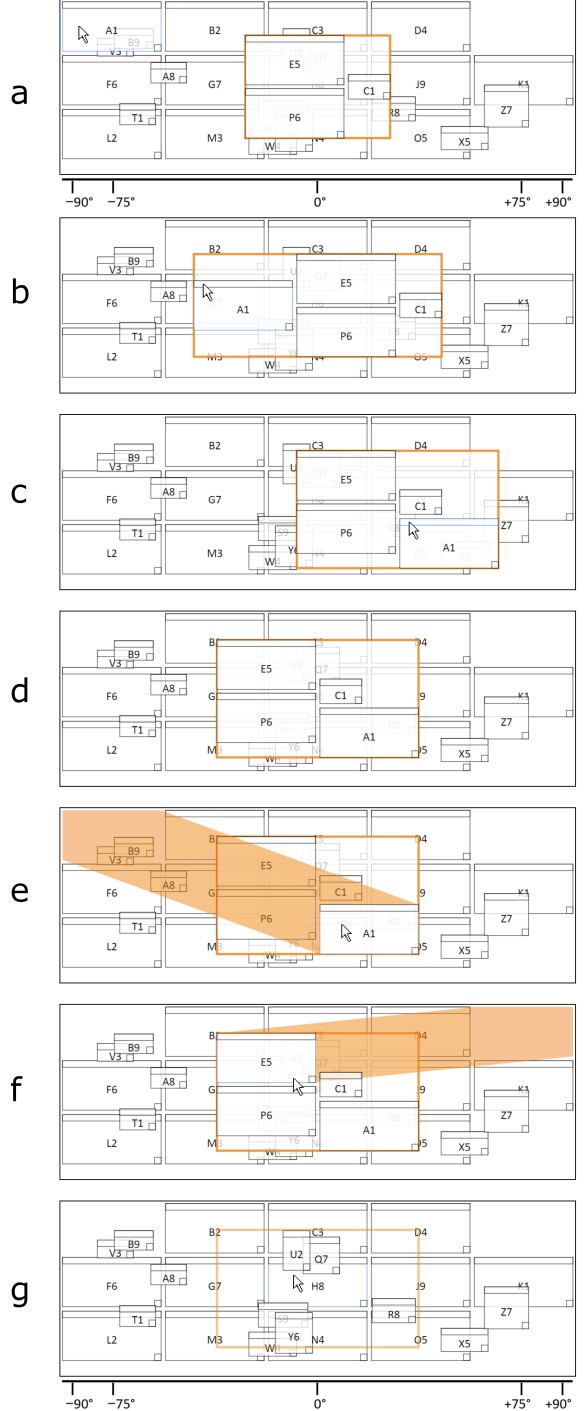

Figure 12: This prototype allows multiple windows to be teleported to the foreground layer, shown in orange.

like the "Windows" key might be more appropriate. When Ctrl is held down, the window manager enters a special mode, within which Alt and the left mouse button activate various functions. Without holding down Ctrl, the other keys and buttons act normally.

Figure 12 illustrates a sequence of interactions in the prototype. The horizontal axis ranges from -93.75 to +93.75 degrees, to match the angle covered by windows in the experiment. Figure 12: (a) The user has placed the mouse cursor over window A1 in upper

left corner. The user presses Ctrl+click and releases to teleport the window to (b) the foreground layer in the center, shown in orange, already containing windows E5, P6, and C1. The user then drags the top of A1, right and downward, to (c) reposition A1 within the foreground. Releasing the mouse button causes the foreground layer to (d) recenter itself. Holding down Ctrl and hovering over windows in the foreground (e and f) causes an orange shadow to show where each window came from in the background. Ctrl-clicking on any window in the background or foreground will teleport it to the foreground, or dismiss it to the background, respectively. Holding down Ctrl+Alt (g) makes the foreground transparent, allowing the user to click on windows behind the foreground to move, resize, or teleport them.

Our prototype displays smoothly animated motion whenever a window is teleported to the foreground, or dismissed to the background, or when the foreground layer automatically recenters itself because one of its windows has been moved, resized, or dismissed. As in the previous prototype, each window remembers its original location in the background when it is dismissed, preserving the layout of the background. When not holding down the Ctrl key, the user is free to reposition and resize any window, within the background or foreground, by grabbing the normal widgets on the window's frame.

Whenever a window is teleported to the foreground for the first time (like in Figure 12(b)), it is automatically given a position within the foreground, either to the left or to the right of the existing foreground windows, depending on whether it came from the left or the right side of the background. Subsequently, windows remember whatever position they had in the foreground. Thus, when the user repositions window A1 in Figure 12(c), if the user were to dismiss and then teleport again A1, A1 would return to the same relative position in Figure 12(c).

## 7 CONCLUSION

The use of teleportation allows a user to retain the benefits of a cockpit layout (e.g., ability to monitor more data than with a single window), but also benefit from reduced head rotation and reduced neck fatigue, as shown in our experiment. Teleportation was also preferred by 9/16 users over the other two conditions in our study.

We subsequently presented a prototype of a window manager that allows multiple windows to be teleported to the center of the user's FoV, with automatic centering, making it easy to dismiss any window to the background, where the original layout of windows is preserved.

## 8 FUTURE WORK

Future work could evaluate our prototype from Section 6, and/or experimentally compare alternative mechanisms for selecting the window to teleport, such as (1) the mouse cursor, (2) eye-gaze [7], (3) a combination of head- and eye-gaze [17], or (4) allowing the user to popup a miniature view of all their windows at the center of their workspace, within which they can select a small proxy of the full-size window to teleport, while using very small eye or neck rotation. This 4th idea is inspired by the "Exposé" feature in macOS which shrinks all windows to temporarily eliminate overlap.

Future work could also apply our set of 4 tasks and vary the 3 parameters described in section 4.4 to evaluate other window management techniques in VR or immersive environments.

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
