# OpenReview forum: "Teleporting Windows to Improve Virtual Reality Cockpits"
_graphicsinterface.org/Graphics_Interface/2023/Conference — Submitted to GI 2023_

### Official Review · Reviewer_tUAw · 2023-01-11
**Nice simple system and study, but contribution is thin**

**Rating:** 5
**Confidence:** 3

**Review:**

This paper presents a somewhat novel interaction technique for managing windows in virtual reality environments. The idea is to teleport a window a central location to minimize the next strain associated with a cockpit layout (effectively a large grid of windows) and to speed up the interactions over a desktop-computing analogous single window layout.

These three approaches were compared for subjective neck strain and window selection time over four different task types. Overall, there seemed to be no advantage of the teleportation technique for window selection time, but there was an improvement on subjective neck strain.

The paper goes on to propose an additional that supports multiple foreground windows (which is not implemented I. VR or evaluated).

On the plus side, the paper is well written and proposes a logical improvement for a real ergonomic issue that will face VR users. The paper claims some novelty with the teleportation technique and acknowledges it’s similarity to other previously proposed interactions.

The main concern I think is with the level of contribution. The technique is extremely simple, which is not always a problem for a paper, but in this case we have something that is approaching “full paper” length and there is very little discussion of the results and the implications of the techniques and the findings. Is teleportation with the effort? Would people use it? Why was it not faster? Even a standard cursory discussion is missing and should really be added.

Additionally, the proposed prototype is compelling, but offers little value to the paper as it is not really considered in the related work. It would have been nice if it was actually implemented and evaluated somehow. I realize this is a big ask, but it seems warranted for a full paper at GI. Otherwise, I believe we have something more akin to a poster level contribution.

I do not mind the non-familiar (to me) data analysis but I found the organization of the experiment and results a little confusing. Breaking apart the metrics from the results and explaining the reasoning for the metrics, in the order that they are presented would be helpful. Figure 7 is not explained in the text as another example.

Overall, I believe that the work has good potential but the current presentation does not provide enough to warrant a paper.

---

### Official Review · Reviewer_71Zq · 2023-01-13
**Credible analysis that ad hoc 3D virtual desktop manager outperforms naive solutions**

**Rating:** 6
**Confidence:** 4

**Review:**

This is a very careful HCI analysis of a somewhat ad-hoc technique for an important use case, with some questions lingering about key aspects of interaction. Overall, the work is acceptable and interesting for GI. I hope that the authors follow up by applying their careful analysis with more control regarding the hand motion and visualization, and to a broader range of virtual desktop management designs.

The way the abstract and paper are written presupposes a somewhat narrow (but really important!) VR application, of virtual desktops. The title and abstract should make clear that this is the target. I picked up the paper thinking that it would be about navigating 3D environments and teleporting the reference frame within them, not about a virtual desktop window manager.

The design choice of head gaze instead of eye gaze or mouse movement is well reasoned and convincing. The overall design of the Teleport interaction is quite reasonable, but also begs the question of whether all of the other 2D virtual desktop management techniques (that are explained in related work) could be applied and which would be best.

The methodology and hypothesis are clearly explained. The choice of particular hotkey controls (such as backtick) is perplexing. How could a user reliably hit somewhat obscure keys when in VR? Is the keyboard's inferred hand position and rendering really sufficient to make the rendered virtual keyboard 1:1? There is no explicit 3D hand tracking in this system. Why not use VR gestures or additional mouse buttons? I find the observations that the users "seemed to remember the mouse position" and thus needed no visual representation and that the monitor selection ray was at 15 degrees below eyeline based on ergonomics recommendations (rather than measurements observations of where users actually look in VR) to be rather cursory for important issues that could affect the results of the study.

The VR display used has a very limited FOV (90 deg) compared to the 150 degree virtual monitor layout. This is potentially a confounding factor as the user must turn their head to even perceive the existence of the side windows instead of relying on peripheral vision.

The explanation of the actual study performed and presentation of results are exemplary.

---

### Official Review · Reviewer_MS7k · 2023-01-13
**Great presentation, limited contribution**

**Rating:** 4
**Confidence:** 4

**Review:**

The authors propose a multi-window arrangement technique that teleports windows to the center of the users viewpoint. This reduces neck movement and consequently neck fatigue, as users are no longer required to look at off-center windows for extended periods of time. They offer a comparative analysis of their Cockpit-with-window-teleportation method, a desktop-style window placement, and a Cockpit-with-fixed-window-placement method (including off-center).

To summarize, this paper presents the cascading windows functionality typically associated with the modifier key e.g. "Windows key", in VR. In the final section, the authors discuss an extension to allow multiple windows in the foreground, although this was only a proof-of-concept implemented in Javascript and not in VR.

I value the effort that the authors put into this project. The paper is straightforward and pertinent to VR. However, the motivation for this study is weak: why is it necessary to conduct a study to determine that moving your neck can cause neck fatigue more so than not moving your neck?

Why did the distance between the windows in the bottom row differ from the distance between the other windows, i.e. 72cm versus 80cm? Why not position the windows at locations sampled from a sphere with an 80cm radius to prevent overlap and occlusion?

In desktop mode, only window titles are displayed and selection is sequential. To select the tenth window, the user must press Alt+Tab ten times, which is slower than expected. Can you provide examples of VR applications that utilise this method of selection?
A fair comparison would display cascading windows (modifier key e.g. Windows or Mac) and permit the user to select with a single click, as is the case with the other conditions. This is the case with the majority of VR applications I've used. Due to the user's familiarity with this method, I would argue that this circumstance would lead to less neck fatigue and a quicker task completion time.


Have you considered other conditions such  be only-vertical, and only-horizontal window placement as in [1]?

[1] Ng, Alexander, et al. "The Passenger Experience of Mixed Reality Virtual Display Layouts in Airplane Environments." 2021 IEEE International Symposium on Mixed and Augmented Reality (ISMAR). IEEE, 2021.



Overall, I find the contribution of this paper to be very limited.

---

### Meta-Review · Area_Chair_wrvv · 2023-01-21

**Recommendation:** 5
**Confidence:** 5

**Metareview:**

The paper is a good presentation of a carefully executed experiment on a relevant topic. The magnitude of the result and thus the contribution of the work are limited by design choices in the construction of the experiment and the narrow scope of the desktop-inspired techniques under investigation.

All reviewers concur with the assessment and suggest concrete ways to extend and repeat the experiment that could lead to a stronger paper, substituting new experiments and results sections into what is already a good written framework for reporting the work. We understand that the experimentation required is beyond the timescale of a revision. We thus recommend rejection at GI but strongly encourage the authors to pursue publication of an improved experiment in this line.